**METHODS AND PROTOCOLS**
Novel Systems Biology Techniques

# manta: a Clustering Algorithm for Weighted Ecological Networks

Lisa Röttjers,[a] Karoline Faust[a]

aLaboratory of Molecular Bacteriology (Rega Institute), Department of Microbiology, Immunology and Transplantation, KU Leuven, Leuven, Belgium

**ABSTRACT** Microbial network inference and analysis have become successful approaches to extract biological hypotheses from microbial sequencing data. Network clustering is a crucial step in this analysis. Here, we present a novel heuristic network clustering algorithm, manta, which clusters nodes in weighted networks. In contrast to existing algorithms, manta exploits negative edges while differentiating between weak and strong cluster assignments. For this reason, manta can tackle gradients and is able to avoid clustering problematic nodes. In addition, manta assesses the robustness of cluster assignment, which makes it more robust to noisy data than most existing tools. On noise-free synthetic data, manta equals or outperforms existing algorithms, while it identifies biologically relevant subcompositions in real-world data sets. On a cheese rind data set, manta identifies groups of taxa that correspond to intermediate moisture content in the rinds, while on an ocean data set, the algorithm identifies a cluster of organisms that were reduced in abundance during a transition period but did not correlate strongly to biochemical parameters that changed during the transition period. These case studies demonstrate the power of manta as a tool that identifies biologically informative groups within microbial networks.

**IMPORTANCE** manta comes with unique strengths, such as the abilities to identify nodes that represent an intermediate between clusters, to exploit negative edges, and to assess the robustness of cluster membership. manta does not require parameter tuning, is straightforward to install and run, and can be easily combined with existing microbial network inference tools.

**KEYWORDS** microbial ecology, network analysis, bioinformatics, clustering, microbiome, networks

As most environmental covariates can explain only a small fraction of the variation in microbial communities, other factors such as species interactions have been suggested to play a large role (1). A range of tools has become available to predict potential interactions as associations. Many of these tools can predict positive as well as negative associations, with the exception of some approaches, such as those based on mutual information (2). Consequently, most microbial networks can be assigned edge weights that quantify the strength of the association. While the exact values of such edge weights may differ depending on tool usage, the sign is highly informative. Microbes can cooccur or exclude each other, as a range of ecological interactions takes place (3). Previous work on ecological networks has demonstrated that the ratios of these interaction types may have implications for biodiversity and ecosystem stability (4–6).

While ecosystem stability, biodiversity patterns, and nestedness have been linked to patterns of biotic interactions (5, 7, 8), such ecological properties cannot be translated to association networks. Unlike many ecological networks, microbial association networks suffer from interpretational challenges, as they cannot be observed directly (9).

This article followed an open peer review process. The review history can be read here.

Address correspondence to Karoline Faust, karoline.faust@kuleuven.be.

Clusters in microbial networks can reveal drivers of community structure. This manuscript describes a novel flow-based network clustering algorithm to identify such clusters and demonstrates how they can be used to identify unique functional groups.

**TABLE 1** Overview of different clustering algorithms[a]

| Algorithm | Cluster no. criterion | Unassigned nodes | No edge filtering required | Preferred network type | No parameter tuning |
|---|---|---|---|---|---|
| manta | Optimize sparsity | Yes | Yes | Undirected | Yes |
| WGCNA signed | Dynamic branch cut | Yes | Yes | None (constructs network) | Yes |
| WGCNA unsigned | Dynamic branch cut | Yes | No | None (constructs network) | Yes |
| Louvain method | Optimize modularity | No | Yes | Undirected; directed version possible | No |
| MCL | Parameter dependent | No | Yes, if parameters optimized | Undirected | No |
| Girvan-Newman algorithm | User dependent | No | No | Undirected; directed version possible | Yes |
| Kernighan-Lin bisection | Only bisection | No | Yes | Any | Yes |

[a]Different properties of manta, WGCNA, MCL, Louvain community detection, the Girvan-Newman algorithm, and the Kernighan-Lin bisection algorithm. The following properties are summarized: how algorithms choose a cluster number, whether they can leave nodes unassigned, whether they perform better with negatively weighted edges removed, and what types of networks they accept. Finally, we assessed whether algorithms required extensive parameter tuning before achieving optimal performance on simulated data.

Moreover, tools used to infer associations generally suffer from high error rates (10). Despite these drawbacks, clusters from microbial association networks have been shown to reflect important drivers of community composition (11, 12). However, traditional choices for network clustering algorithms are unable to make optimal use of information contained in edge signs. For example, the Markov cluster algorithm (MCL) uses a probability matrix to identify clusters (13). While a weighted adjacency matrix can be scaled to generate a probability matrix, the algorithm depends on edge density to infer clusters and is therefore mostly suitable for networks with a low number of negatively weighted intercluster edges.

Alternatives that are able to take edge weight into account are available, such as the Louvain method for community detection (14) and the Kernighan-Lin bisection algorithm (15), an algorithm that optimizes separation of the network into two parts. A different approach is to scale node weights such that negatively weighted edges are either converted to positively weighted edges or have lower positive weights; this approach is implemented in WGCNA (16), a pipeline for network inference and clustering. Although scaling approaches result in a loss of sign information, such approaches can also yield satisfactory results.

While some of these clustering approaches can cluster weighted networks, they have not specifically been designed to detect association patterns generated through ecological processes. Even if microorganisms are not directly interacting, they may cooccur as a result of niche filtering (17), or they may actively prevent establishment of other species. Such antagonistic relationships have been described for sponge symbionts (18), the plant rhizosphere (19), and biofilm formation (20, 21). Regardless of the ongoing process, cooccurrence with one species but not another can be described as "the enemy of my enemy is my friend" in an undirected network. Even if there are no direct interactions between cooccurring species, a powerful shared negative association introduced through niche filtering or competition may be sufficient to result in observed cooccurrence.

In this work, we describe manta (microbial association network clustering algorithm), a novel method for clustering microbial networks. We demonstrate in two case studies that our method can identify subcompositions with biological importance that cannot be found through conventional correlation strategies. Moreover, we include a robustness metric to quantify whether cluster assignments are robust to the high error rates found in microbial association networks. Our method represents an alternative to the popular MCL algorithm (13) that, in contrast to MCL, can take advantage of edge signs and does not need parameter optimization (Table 1).

(This article was submitted to an online preprint archive [22].)

## RESULTS

**manta implements features for clustering weighted and noisy networks.** The indirect effect of one node on another node in the network can be estimated by

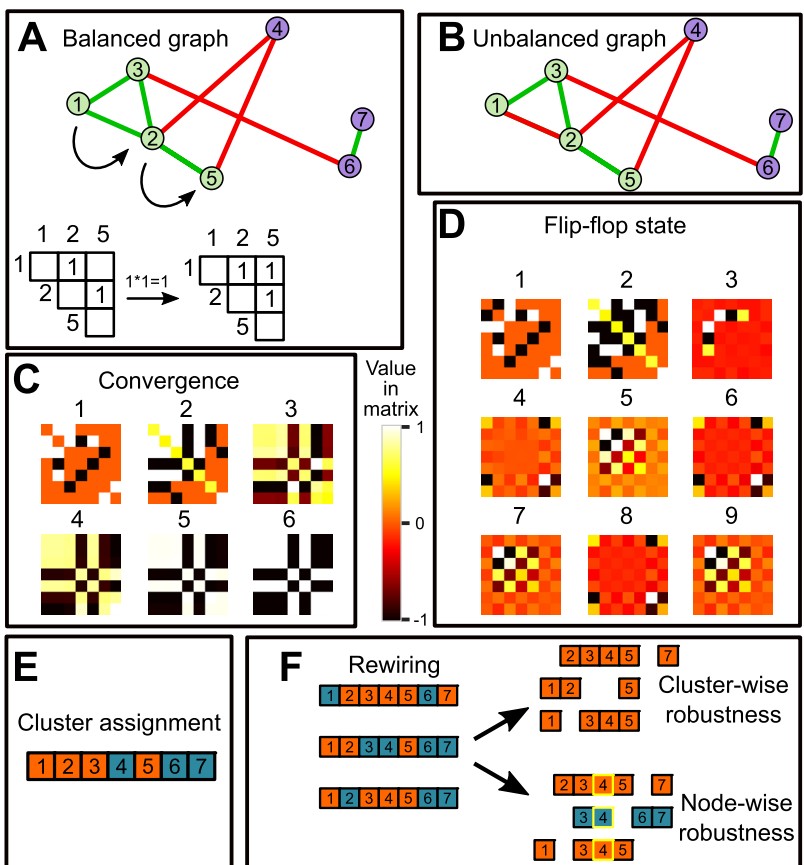

**FIG 1** manta pipeline. (A) Toy graph with two clusters separated by negatively weighted edges. The effect of node *x* on node *z* can be estimated by taking the product of edges 1,2 and 2,5. (B) Toy graph with a single negatively weighted edge in the left cluster. (C) Scoring matrix for panel A across six iterations. Black and white values reflect −1 and 1, respectively. After six iterations, the scoring matrix reaches convergence. (D) Scoring matrix for panel B across nine iterations. Unlike panel C, this matrix reaches a flip-flop state, where the scoring matrix alternates between the configurations shown in iterations 6, 7, 8, and 9. A few values in the matrix reach −1 or 1, while all other values oscillate near 0. (E) manta uses agglomerative clustering on the scoring matrix to assign each node to a cluster. For flip-flopping matrices, the scoring matrix is generated from subsets of the complete network. (F) A fraction of the original network is rewired to generate permuted cluster assignments with identical degree distributions. The robustness of cluster assignments can then be estimated by comparing the Jaccard similarities of cluster memberships cluster-wise or node-wise.

multiplying the weights of the edges connecting the two nodes (Fig. 1A). If the nodes are connected only by positively weighted edges, the indirect effect is also positive. In contrast, if the path between the two nodes contains a single negatively weighted edge, the indirect effect is negative; hence, clusters found by manta reflect the principle "the enemy of my enemy is my friend." Depending on the structure of the network (Fig. 1A and B), manta uses two alternative strategies to generate scoring matrices (Fig. 1C and D). See Materials and Methods for a detailed explanation and the pseudocode describing the algorithm.

After the scoring matrix is generated, it can be clustered with an agglomerative clustering approach (Fig. 1E). The optimal cluster number is identified with the sparsity score (equation 1 in Materials and Methods), which is calculated from intracluster to intercluster weighted edges. The network can then be rewired and the procedure repeated to generate robustness scores (Fig. 1F). This approach generates biologically relevant clusters while ignoring nodes that cannot be confidently assigned to a cluster.

**manta equals or outperforms other algorithms on synthetic data sets.** To evaluate the performance of manta in comparison to alternative methods, we generated synthetic data sets using two different approaches. One is based on the gener-

alized Lotka-Volterra (gLV) equation, while the other (FABIA) was developed for the evaluation of biclustering applied to gene expression data (23). We chose to use these approaches because they provide a ground truth and have entirely different network topologies. For example, the median approximated node connectivity of the gLV networks is 1, in contrast to 46 for the FABIA networks. The latter networks more closely represent ecosystems without biotic interactions, where network topology is governed solely by niche filtering or other dynamics.

Cluster assignments were evaluated by the complex-wise sensitivity (Sn), the cluster-wise positive predictive value (PPV), geometrical accuracy (Acc), and separation (Sep) (24). Additionally, we included the sparsity score used by manta (equation 1) to visualize the ratio of intracluster to intercluster weighted edges. The complex-wise sensitivity estimates the coverage of a true-positive cluster by its best-matching assigned cluster, whereas the cluster-wise positive predictive value measures how well an assigned cluster covers its best-matching true-positive cluster. In contrast, the separation is calculated by taking the product of the fraction of assigned nodes in the true positive clusters and the fraction of true-positive nodes in the assigned clusters. Hence, the separation penalizes for cluster overlap, unlike the reported Acc, PPV, and Sn. A perfect cluster assignment would therefore have a separation score of 1, while worse assignments would reach 0.5 or even lower.

This approach uses the contingency matrix rather than a list of true positives, effectively permitting evaluation of assignments that do not necessarily match the true-positive clusters (see Fig. S1) in the supplemental material. However, these measures can be skewed by cluster assignments that mostly assign all nodes to one cluster (Fig. S1B) or assign almost every node to its own cluster. Additionally, overlapping clusters can inflate some measures of performance (Fig. S1C). To resolve these pathological cases, we filtered assignments where over 80% of the nodes were assigned to a single cluster or over 50 clusters were identified. While Acc, PPV, and Sn can be high for algorithms that assign true-positive clusters to a single cluster, separation is calculated by multiplying the proportion of true-positive nodes in the assigned cluster with the proportion of cluster nodes in the true-positive cluster. Hence, the separation measure punishes cluster assignments that mix up multiple true-positive clusters.

Under the right circumstances, multiple algorithms achieve a separation around 0.6 when clustering the two-cluster network (Fig. 2). However, this requires the use of a positive-edge-only subnetwork for the Louvain method, the WGCNA unsigned approach (data not shown), and the Girvan-Newman algorithm; on the complete network, these algorithms do not separate the nodes into two clusters or fail to separate the true-positive clusters. Shifting the edge weights failed to resolve this (Fig. S9). For MCL, separation depends on the parameter settings (Fig. S2). This algorithm can cluster the complete network but only if the inflation parameter is set to 3 or to another uneven value. In this simulation, MCL with default parameters was unable to recover the true-positive cluster assignment (data not shown). Strikingly, performance is improved for the Louvain method but not for the Kernighan-Lin bisection when only positively weighted edges are considered. While the Louvain method takes edge sign into account during its optimization, negatively weighted edges appear to have a negative effect on separation.

The environmentally motivated clusters from the gLV simulation have an underlying interaction network that can affect network topology. In contrast, the FABIA approach does not have an interaction network from which abundances are generated (Fig. 3). As a result of this change in topology, no algorithm is able to generate cluster assignments that have sparsity scores close to 1. However, not all algorithms are equally affected by the change in topology. Especially affected are those networks that assume a modular or scale-free structure, such as WGCNA. The reduced performance may be attributed to one of WGCNA's core assumptions: gene regulatory networks are assumed to be scale-free. Consequently, WGCNA infers a correlation network and soft-thresholds this network by choosing the matrix power such that scale-freeness is optimized. The change in network topology appears to rescue performance for the unsigned WGCNA

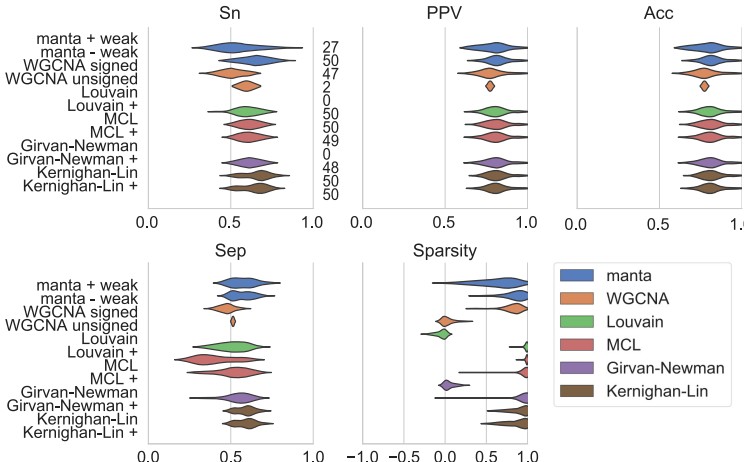

**FIG 2** Performance of network clustering tools on two environmentally motivated clusters. Clustering performance was estimated on 50 independently generated data sets generated from random interaction matrices. Sensitivity (Sn), positive predictive values (PPV), accuracy (Acc), and separation (Sep) were calculated as described previously (24). The sparsity of the assignment is a function of the edge weights of intracluster versus intercluster edges (equation 1). The numbers next to the sensitivity results indicate how many cluster assignments met the following criteria for a particular algorithm: no cluster should exceed 80% of the total number of nodes, and there should be fewer than 50 clusters. The manta algorithm was run with and without weak assignments, while WGCNA was run with signed networks and a signed topological overlap matrix and with unsigned networks combined with the unsigned matrix. For all other algorithms, we provided the complete network in addition to the positive-edge-only network (indicated by +).

method; while this method did not return cluster assignments on the gLV simulated data, it does return cluster assignments on the FABIA simulation. Surprisingly, the opposite result holds for MCL, as the algorithm can no longer return cluster assignments on the positive-edge-only networks. MCL does better on the FABIA networks

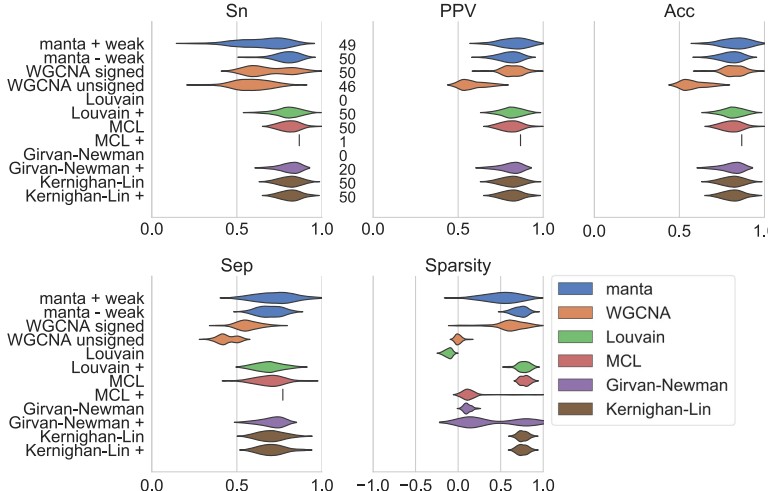

**FIG 3** Performance of network clustering tools on two biclusters generated with FABIA (23). Clustering performance was estimated on 50 independently generated data sets without an underlying topology. Sensitivity (Sn), positive predictive values (PPV), accuracy (Acc), and separation (Sep) were calculated as described previously (24). The sparsity of the assignment is a function of the edge weights of intracluster versus intercluster edges (equation 1). The numbers next to the sensitivity results indicate how many cluster assignments met the following criteria for a particular algorithm: no cluster should exceed 80% of the total number of nodes, and there should be fewer than 50 clusters. The manta algorithm was run with and without weak assignments, while WGCNA was run with signed networks and a signed topological overlap matrix and with unsigned networks combined with the unsigned matrix. For all other algorithms, we provided the complete network in addition to the positive-edge-only network (indicated by +).

with the negatively weighted edges included, while it performed worse on the gLV networks when these were included. This demonstrates that MCL performance can require parameter optimization and may benefit from different preprocessing strategies depending on the network topology.

Neither manta nor the Kernighan-Lin method optimizes toward a modular or scale-free network. As with the environmentally motivated simulation, Kernighan-Lin bisection achieves some of the best results on this simulation. Hence, if users suspect based on a preliminary analysis that their data set contains only two clusters, this algorithm is likely to recover that separation regardless of the underlying structure. However, manta has the advantage of a tunable weak cluster assignment that can handle noisier data and less accurate networks, as well as data with more than two clusters.

While accuracy could not be estimated well on three clusters, manta was able to return cluster assignments with the highest sparsity (Fig. S6) and returned assignments with high separation in increasingly permuted data (Fig. S7). Performance dropped on networks generated from data with added multinomial noise, even if the overall network structure appeared to be preserved (Fig. S8). Additionally, scaling the correlations instead of removing negatively weighted edges failed to improve performance, as most algorithms quit returning cluster assignments or the assignments did not recover the original network structure (Fig. S9).

**manta identifies biologically relevant groups in cheese rinds.** We demonstrate the real-world applicability of manta on a cheese rind data set generated by Wolfe et al. (25). In this study, the authors analyzed 137 cheese rind communities and identified important community members. Moreover, they found that community assembly of cheese rind communities was highly reproducible, despite the large geographical distances between cheeses. This can be explained at least partially by manipulation of the rind biofilm, as cheesemakers can introduce an initial community through starter cultures and then control the environment during the aging process. In their study on cheese rinds, Wolfe et al. (25) originally demonstrated that most of the community variation could be explained by the rind type. Indeed, most samples appear to cluster by rind type (Fig. 4B). The authors found that samples from washed cheeses could cluster closely with both other types of rinds; the principal-coordinate analysis (PCoA) also captures this phenomenon, as samples from washed cheeses are dispersed across the entirety of the axes.

We ran manta on the association network to assess whether cluster analysis would be able to recapitulate some of the drivers of community structure in the cheeses (Fig. 4A). The network visualization of the data reveals some interesting trends, as the network contains three clusters that correlate with moisture and taxonomy. Cluster 1 is mostly comprised of *Firmicutes*, and its summed abundances have a strong negative correlation with moisture. In fact, several of these taxa belong to the genus *Staphylococcus*, replicating the results by Wolfe et al. (25), as they demonstrated that *Staphylococcus* spp. are abundant on dry natural rinds. In contrast, cluster 1 consists mostly of *Proteobacteria* and correlates positively with moisture.

While the clusters correspond well to the results obtained by Wolfe et al. (25), manta is able to offer additional insight into community structure through its separation of *Actinobacteria*. Wolfe et al. (25) demonstrated that the abundance of taxa belonging to this phylum is negatively correlated with moisture. However, the clusters indicate that this correlation is more nuanced. Some of the taxon abundances may reflect a gradual response to moisture rather than a strict preference for dry or moist cheese rinds, as summed abundances for *Actinobacteria* belonging to clusters 2 and 3 display a weak and nonsignificant correlation with moisture rather than the strong negative correlation associated with cluster 1.

On this data set, manta identifies clusters that correspond well with the main drivers of community composition in this study, while also identifying taxa that display

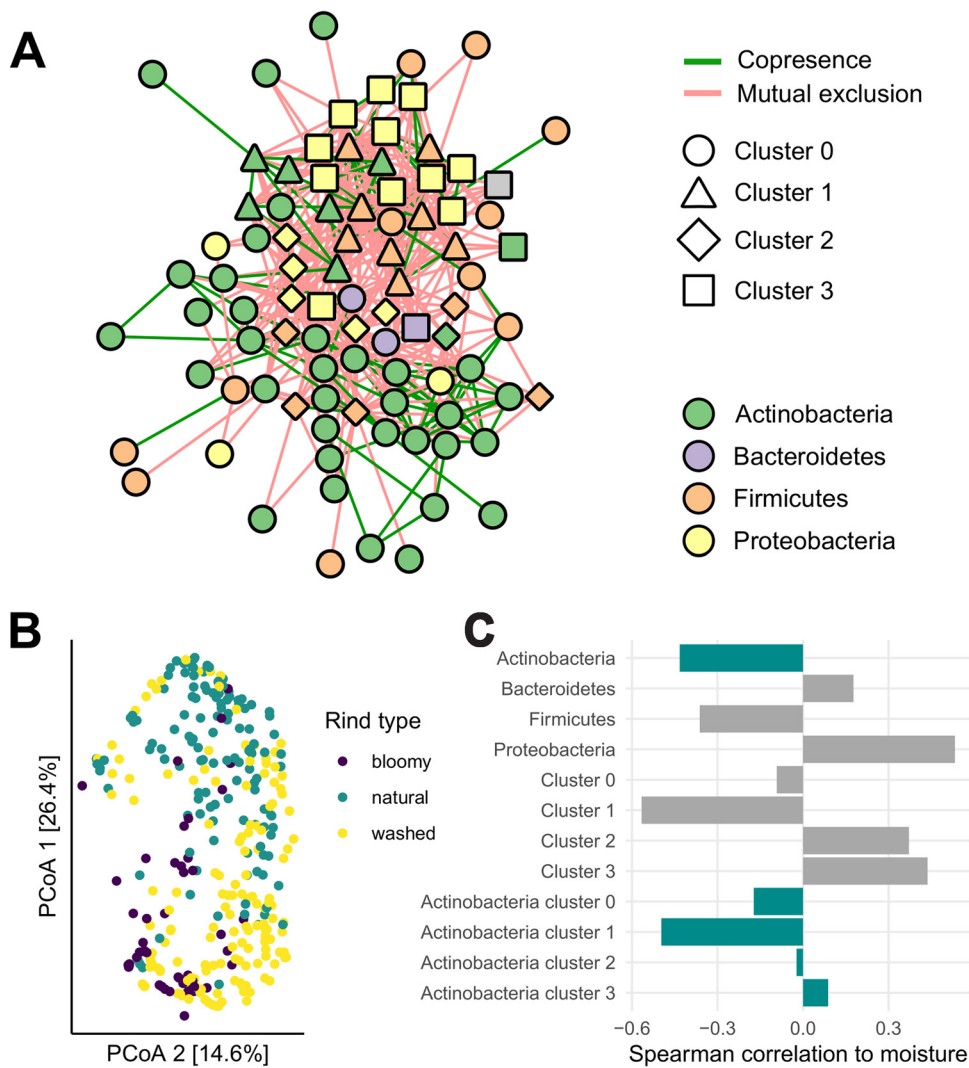

**FIG 4** Network analysis of a cheese data set (25). (A) CoNet network clustered with manta. Cluster identity is represented by node shape, whereas phylum membership is indicated by node color. The edge color is mapped to the sign of the association. (B) Principal-coordinate analysis (PCoA) of Bray-Curtis dissimilarities for sample compositions. The colors indicate the different cheese rind types: bloomy cheeses are inoculated with fungi, while washed cheeses are repeatedly washed with a brine solution. In contrast, natural rinds are not disturbed during aging. (C) Spearman correlation of moisture to summed taxon abundances. Correlations for the phylum *Actinobacteria* are highlighted in blue.

intermediate responses to these drivers. Hence, this case study demonstrates that cluster analysis can yield novel insights into community structure.

**manta detects global trends in coastal plankton communities.** One advantage of manta is its ability to handle networks generated through any type of inference algorithm (though conversion to undirected networks is necessary in some cases). We demonstrate this through a time series analysis of coastal plankton communities (26). Martin-Platero et al. (26) collected samples for 93 days and used these data to demonstrate that the communities changed rapidly, but only when lower taxonomic ranks were taken into account. The authors demonstrated the use of WaveClust on this data set, a novel clustering method based on wavelet analysis of longitudinal data. Consequently, WaveClust can find taxon associations at low and high frequencies that are not visible without a frequency decomposition. The authors evaluated their technique on data from coastal plankton, where clusters identified by WaveClust corresponded to rapid growth of specific groups of taxa. A network analysis of the Granger causalities between clusters and metadata further demonstrated that the clusters could be

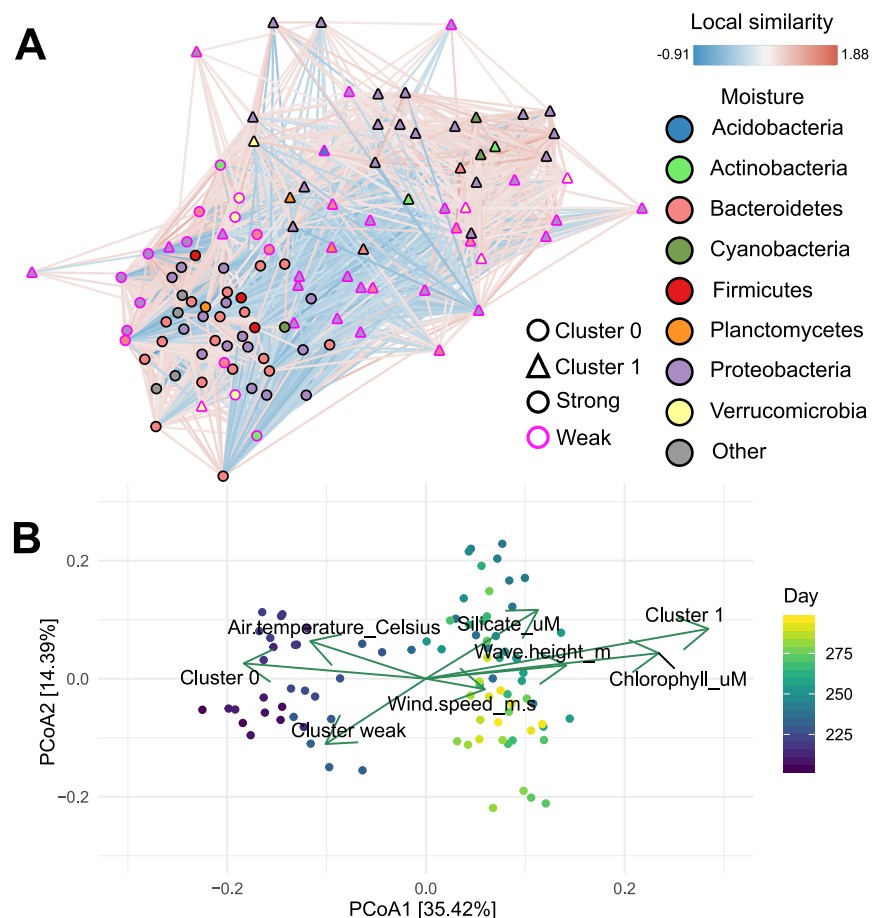

**FIG 5** Network analysis of longitudinal 16S data collected from coastal plankton (26). (A) eLSA network clustered with manta. Cluster identity is represented by node shape, whereas phylum membership is indicated by node color. The edge color is mapped to the local similarity score. Nodes that were not assigned to a cluster by manta are shown with a pink border. (B) Principal-coordinate analysis (PCoA) of Bray-Curtis dissimilarities for sample compositions, overlaid with significantly covarying environmental vectors and cluster abundance vectors. The significance of these vectors was assessed through permutation testing; only significant vectors are shown. Cluster abundance vectors were scaled independently from environmental vectors; abundances of taxa that could not be assigned to a cluster were included in the "Cluster weak" vector. The axis values are the eigenvalues of the ordination axes.

separated into two regimes, both corresponding with an initial warm period followed by a rapid or gradual cooling period. These regimes are also visible in the ordination plot, as the samples separate over time around day 215 (Fig. 5B). We generated a network with eLSA and clustered it with manta (Fig. 5A).

manta is not designed for time series and therefore is not able to detect frequency-specific associations. However, manta does summarize larger trends in the data. To demonstrate this, we first computed a network with eLSA, a microbial network inference algorithm that takes temporal shifts into account (27, 28). A force-directed layout reveals that the eLSA network contains a large number of anticorrelated nodes. These clusters correlate with multiple environmental variables, thereby closely reflecting the two metaregimes identified by Martin-Platero et al. (26). The vector for abundances of taxa belonging to cluster 1 aligns closely with the chlorophyll concentration and wave height; in contrast, the total abundance of taxa belonging to cluster 0 was highest at the start of the experiment, when the seasonal warm period was still ongoing. The vector for taxa that could not confidently be assigned to a cluster was also significantly covarying with community composition and is directly opposite of the silicate concentration. The silicate concentration is not part of the originally reported Granger causality model (26), but it does increase sharply around the transition point between the two

metaregimes (days 240 to 260). Hence, the weak assignments capture a set of taxa that become less abundant during the transition period and do not correlate strongly to taxa in the assigned clusters.

## DISCUSSION

The manta algorithm is able to perform as well as or better than preexisting algorithms available for network clustering, while its ability to identify weak cluster assignments can assist in defining cluster cores in networks that represent a gradient rather than distinctly separated clusters. Additionally, since it has been designed specifically to cluster networks with negatively weighted edges, it is better able to exploit this information than several alternative algorithms. Our case studies demonstrate how cluster assignments recapitulate main drivers of community composition.

A key limitation of manta is its inability to deal with networks with only a few negative edges. In such cases, cluster assignments mostly separate central nodes from peripheral nodes. However, manta was specifically developed for weighted networks; when only a few edges have negative weights, treating the network as unweighted or removing negatively weighted edges will not affect network structure significantly. As demonstrated by the separation scores on positive-edge-only networks, algorithms like MCL and the Louvain method will then perform adequately. Moreover, the clusters assigned by manta can accurately be described as "the enemy of my enemy is my friend"; while there are algorithms available that require nodes to be similar to nodes within the community (29), manta has no such requirements and may assign nodes to a cluster even though they are not necessarily positively associated with other nodes within that cluster. Additionally, users should be aware that cluster assignments will reflect only main drivers of community composition, as manta tends to generate a small number of clusters separated by weakly assigned nodes. Separating a data set by its main drivers (e.g., sample type or location) can help identify more interesting clusters.

Although we chose to evaluate manta in the context of microbial networks in this paper, manta may also be useful for clustering other types of networks with a large number of negative edges. For example, even though WGCNA was originally developed for gene expression data (16), it has also been used for microbial data (11). The ability of manta to identify clusters without any underlying topological structure implies that it may be especially valuable in contexts where a small-world or scale-free structure cannot be expected. Moreover, its lack of sensitivity to parameter settings in this simulation demonstrates that manta is applicable in situations where little is known about the structure of the analyzed network.

## MATERIALS AND METHODS

Unless otherwise specified, computations were carried out in R v3.5.1 and Python v3.6.3. Correlation networks were generated with the rcorr function from the Hmisc R library (version 4.2-0) (30). Analysis of simulated data sets was carried out in Python using NetworkX (version 2.1) (31), numpy (version 1.15.4) (32), pandas (version 0.21.0) (33), and scipy (version 1.2.0) (34). Additional analyses for case studies were carried out in R using igraph (version 1.2.4.1) (35), phyloseq (version 1.26.1) (36), and vegan (version 2.5-5) (37). In addition to the specified versions of NetworkX, numpy, and scipy, manta uses scikit-learn (version 0.19.1) (38).

**Clustering by graph traversal.** We can represent a graph as an adjacency matrix, with each node in the graph represented as a row and column in the matrix. A nonzero entry in row $i$ and column $j$ represents an edge between node $i$ and node $j$. In the case of an undirected graph, the adjacency matrix is symmetric, so position $i$, $j$ is identical to position $j$, $i$. The weighted adjacency matrix, where the values in the matrix are the edge weights, is used by manta to cluster networks without losing sign information.

manta was originally designed as an alternative to MCL, which uses random walks to find clusters. MCL does so through iterations of matrix expansion and inflation so that values in the matrix eventually converge to 0 or 1. The process of modeling random walks as Markov chains requires the adjacency matrix to be converted to a stochastic matrix in each iteration, so that all columns sum to 1. This normalization step removes the signs from all values. Since manta uses an alternative normalization method to retain the signs, manta does not model Markov chains. The pseudocode below describes the core algorithm (Tables 2 to 4). The adapted normalization steps (Table 2) and inflation steps (Table 3, steps 9 to 11) retain the signs introduced by the weighted adjacency matrix. We also tested other variants of these steps, i.e., raising each element to a power or taking the root; unlike the current inflation step, no variant was able to capture the cluster structure. While this step may seem counterintuitive, as it

**TABLE 2** Pseudocode for normalizing scoring matrix (algorithm 1)[a]

| Step | Description |
|------|-------------|
| 1 | $M \leftarrow$ scoring matrix |
| 2 | **for** row, column in $M$ **do** |
| 3 | $M_{\text{row,column}} \leftarrow M_{\text{row,column}}/\max[\text{abs}(M)]$ |
| 4 | **end** |

[a]Pseudocode operators "for," "do," and "end" are highlighted in bold for clarity.

reduces the difference between small and large values in the matrix (rather than inflating them as in MCL), values converge to −1 and 1 each time for the toy model (Fig. 1B).

Like MCL, the time and space complexity of manta is a function of the matrix dimensions. The most time-intensive operation is the matrix power: for a $n$ times $n$ matrix, the time complexity of a naive matrix power is $O(n^3)$ (39). MCL has tackled this problem through a pruning strategy that removes values close to zero, which reduces the time complexity of the algorithm. However, the complexity of the manta intermediate matrices increases rather than decreases as each position converges toward –1 or 1, so no pruning strategy could be implemented. Since manta uses numpy to carry out matrix multiplications, the time complexity is instead determined by high-performance BLAS implementations and is a bit lower than $O(n^3)$ in practice (40). As the entire matrix is stored in memory, the space complexity is $O(n^2)$. The time and space complexities provided above ignore the number of iterations; generally, between 50 and 100 iterations are sufficient to generate a scoring matrix for an unbalanced graph, while a balanced graph will need even fewer. On the 100-species gLV example, the complete algorithm took approximately 7 seconds to run (Windows, Intel Xeon E3-1505M processor, 3.00 GHz, 32 GB memory).

As described by Van Dongen (13), classes of matrices that do not converge to a stable state after repeated iterations exist. Instead, these matrices exhibit flip-flop equilibrium states and switch to alternative configurations with each iteration. While these flip-flop states represent rare cases when MCL is applied, the use of signed graphs by manta strongly increases the probability of these states to appear. This relates to the notion of balance in signed graphs (41), where a graph is balanced only if the product of edge weights in every cycle is positive. A cycle is a closed chain of nodes; for example, the cycle 1, 2, 3 in the balanced toy model (Fig. 1A) has a positive product, while the same cycle in the unbalanced toy model (Fig. 1B) has a negative product. The balance of the graph matters for the expansion step; we observed that the sign of nonzero elements of the expanded scoring matrix never conflicts with the sign of nonzero elements in the weighted adjacency matrix if the adjacency matrix corresponds to a balanced graph.

If the graph is not balanced (42), manta carries out the previously described diffusion procedure on a subset of nodes (Table 4). Only one iteration is carried out on this subset, as any more iterations would lead to the appearance of flip-flop states. The subsetting approach is complemented by an analysis of any balanced components in the graph; if those are present, multiple iterations of expansion and inflation are carried out on the balanced subgraph until convergence occurs. The scoring matrix used for cluster assignment is reconstructed from the subsets, but only from positions in the subsets where the sign of the value is consistent with the sign reported by most subsets. However, the accumulation of high

**TABLE 3** Pseudocode for balanced graphs (algorithm 2)[a]

| Step | Description |
|------|-------------|
| 1 | $M[0] \leftarrow$ scoring matrix initialized from weighted adjacency matrix |
| 2 | $\epsilon \leftarrow$ threshold for convergence |
| 3 | $E \leftarrow$ initial error value, set to 1 |
| 4 | $I \leftarrow$ maximum number of iterations |
| 5 | **for** $i = 1$ to $I$ **do** |
| 6 | **while** $100E > \epsilon$ **then** |
| 7 | $M[i] \leftarrow M[i-1]^2$ |
| 8 | $M[i] \leftarrow$ apply algorithm 1 on $M[i]$ |
| 9 | **for** row, column in $M[i]$ **do** |
| 10 | $M[i]_{\text{row,column}} \leftarrow M[i]_{\text{row,column}} + 1/M[i]_{\text{row,column}}$ |
| 11 | **end** |
| 12 | $M[i] \leftarrow$ apply algorithm 1 on $M[i]$ |
| 13 | $E \leftarrow 0$ |
| 14 | $b \leftarrow 0$ |
| 15 | **for** row, column in $M[i]$ **do** |
| 16 | **if** $M[i]_{\text{row,column}}$ **is not** 0 **then** |
| 17 | $b \leftarrow b+1$ |
| 18 | $E \leftarrow M[i]_{\text{row,column}} - M[i-1]_{\text{row,column}}$ |
| 19 | **end** |
| 20 | $E \leftarrow E/b$ |
| 21 | **end** |

[a]Pseudocode operators "while," "for," "if," "is not," "then," "do," and "end" are highlighted in bold for clarity.

**TABLE 4** Pseudocode for unbalanced graphs (algorithm 3)[a]

| Step | Description |
|------|-------------|
| 1 | $G \leftarrow$ original graph |
| 2 | $K \leftarrow$ no. of subgraphs to generate |
| 3 | $F \leftarrow$ no. of edges to subsample |
| 4 | $R \leftarrow$ fraction of subgraphs where edges need to be stable |
| 5 | **for** $k = 1$ to $K$ **do** |
| 6 | sample $F$ edges from $G$ to create $g$ |
| 7 | **if** components in $g$ are balanced **then** |
| 8 | **for** component in $g$ **do** |
| 9 | $B\leftarrow$ apply algorithm 2 on component |
| 10 | **end** |
| 11 | $M[k]\leftarrow$ scoring matrix initialized from $g$ |
| 12 | $M[k]\leftarrow M[k]^2$ |
| 13 | $M[k]\leftarrow$ apply algorithm 1 on $M[k]$ |
| 14 | **for** row, column in $M[k]$ **do** |
| 15 | $M[k]_{\mathrm{row,column}}\leftarrow M[k]_{\mathrm{row,column}} + 1/M[k]_{\mathrm{row,column}}$ |
| 16 | **end** |
| 17 | $M[k]\leftarrow$ apply algorithm 1 on $M[k]$ |
| 18 | $M[k]\leftarrow M[k] + B$ |
| 19 | $M[k]\leftarrow$ apply algorithm 1 on $M[k]$ |
| 20 | **end** |
| 21 | $O \leftarrow$ empty matrix |
| 22 | **for** row, column in $M$ **do** |
| 23 | $p\leftarrow 0$ |
| 24 | $n\leftarrow 0$ |
| 25 | **for** $k = 1$ to $K$ **do** |
| 26 | **if** $M[k]_{\mathrm{row,column}}>0$ **then** |
| 27 | $p\leftarrow p + 1$ |
| 28 | **else if** $M[k]_{\mathrm{row,column}}<0$ **then** |
| 29 | $n\leftarrow n + 1$ |
| 30 | **if** $p/K>R$ **or if** $n/K>R$ **then** |
| 31 | **for** $k = 1$ to $K$ **do** |
| 32 | $O_{\mathrm{row,column}}\leftarrow O_{\mathrm{row,column}} + M[k]_{\mathrm{row,column}}$ |
| 33 | **end** |
| 34 | **end** |
| 35 | $O\leftarrow O/\max[\mathrm{abs}(O)]$ |

[a]Pseudocode operators "while," "for," "if," "or," "else," "then," "do," and "end" are highlighted in bold for clarity.

scores for central nodes prevents the agglomerative clustering algorithm from identifying relevant clusters. Therefore, whenever small clusters with a size below a user-specified threshold are detected, rows and columns in the scoring matrix that correspond to these small clusters are removed and clusters are calculated from the remaining values. The removed nodes are then assigned to a cluster based on the average shortest path weights to cluster members.

With the scoring matrix $M$ as input, we use agglomerative clustering on Euclidean distances with Ward's minimum variance method as a linkage function to define clusters (38). The optimization criterion for choosing the optimal cluster number is based on the number of cut edges (equation 1). A graph $G$ can be defined as $G = (V, e)$, with $v$ and $e$, respectively, comprising the vertices and edges that make up the graph. For such a graph with edge weights $w$, the sparsity score of cluster assignments is calculated as a function of the sum of the cardinalities of different sets of edges based on their cluster assignment (equation 1). Consequently, the sparsity score ranges from $-1$ to $1$, with $-1$ being the worst possible cluster assignment: in this case, all negatively weighted edges are placed inside clusters and all positively weighted edges are placed outside clusters.

$$s = \frac{1}{|N|}\left[|e(w < 0) \notin C| + |e(w > 0) \in C|\right.$$
$$\left. - |e(w < 0) \in C| - |e(w > 0) \notin C|\right] \tag{1}$$

where $s$ is the sparsity score, $e$ is an edge, $w$ is the edge weight, $C$ is the set of edges inside clusters, and $N$ is the total set of edges in the graph.

manta is in theory able to handle directed graphs; in practice, strict limitations apply, and the algorithm usually converges to zero.

**Weak assignments and robustness.** The subsetting strategy is complemented by additional information generated through flip-flopping iterations of the network. The limits of the scoring matrix approach $-1$ and $1$ during flip-flop iterations, but these limits are approached only by a few diagonal values; in a balanced matrix, all diagonal values should approach 1 (Fig. 1C). We identified a unique set of nodes corresponding to these diagonals: oscillators. The maximum of the diagonal in the scoring matrix approaches the positive limit for these nodes, while one position in $M$ in the same row/column

as the oscillator reaches the minimum. With the oscillators, manta can identify the shortest paths that assess whether nodes have edges that are in conflict with their cluster assignment. For each shortest path $n$ from node $v$ to oscillator $t$, the product $\delta$ of the scaled edge weights is calculated. The average of these products is then the mean edge product (equation 2).

$$P_{v,t} = \frac{1}{|n|} \sum_{i=1}^{n} \delta_i(v, t) \qquad (2)$$

where $P$ is the mean edge product, $v$ is a node, $t$ is an oscillator, $n$ is the set of shortest paths from $v$ to $t$, and $\delta$ is the product of weights from one shortest path $i$.

The node is considered to have a weak assignment if $P$ meets one of the following criteria: (i) the sign of $P_{v,t}$ is not positive for $v$ assigned to the same cluster as $t$ or (ii) $P$ is smaller than a user-specified threshold.

The weak assignment does not check for nodes that belong to two clusters but rather filters out nodes that could not be assigned to any cluster, given that the shortest paths were in conflict with the node's positions in the scoring matrix.

In addition to the identification of weakly assigned nodes, we developed two new robustness measures to identify robust node assignments and cluster compositions (Fig. 1E). These measures were inspired by the reliability metric developed by Frantz and Carley (43) and highlight parts of the clustering outcome that are disproportionately sensitive to errors in network inference. Both measures are generated from rewired networks and are reported as a confidence interval of Jaccard similarity coefficients. For the cluster-wise robustness, Jaccard similarity scores are computed for the original clusters and their best permuted matches (identified through the maximum Jaccard similarity coefficient).

These confidence intervals provide two different types of information. First, a low Jaccard similarity coefficient indicates that a cluster has a different composition given a few errors or that a node is assigned to a cluster with an entirely different composition given a few errors. Second, the width of the confidence interval demonstrates how variable cluster assignments are; wide intervals for node-wise robustness demonstrate that the node sometimes ends up in a similar cluster but can also be assigned to a different cluster.

**Synthetic data sets.** We carried out two types of simulations with 50 replicates per simulation. For the first type, species interaction networks with a connectivity of 5% were generated with the R package seqtime (version 0.1.1). The effects of environmental factors on growth rates were sampled from a normal distribution with $\mu = 1$. The strengths of these factors were sampled from a normal distribution with $\mu = 3$. To ensure that the cumulative effects of the environmental factors were unique to each condition, one factor was set to be positively weighted while the remaining two were converted to negative values. Data sets were then generated with the generalized Lotka-Volterra equation, with growth rates of each organism adjusted per environmental condition. As some interaction matrices caused population explosions, these were regenerated until enough matrices were available for which the generalized Lotka-Volterra equation could be solved.

For details and source code on these data sets, see reference 9; note that in contrast to this work, we did not enforce a scale-free structure in the interaction network. We generated densely connected networks from the simulated species abundances with the Pearson correlation and filtered these for significance ($\alpha = 0.05$).

The second type of simulation uses an adapted version of the FABIA R package (23), a package that generates simulated data for evaluations of biclustering algorithms. We adapted the makeFabiaDatablocksPos function to generate biclusters at set locations and used these locations to define true-positive clusters. The constructed matrix was then used to infer a correlation network in the same manner as the gLV networks.

**Clustering algorithms.** We compared manta to the Louvain method (14), MCL (13), WGCNA (16), the Girvan-Newman method (44), and Kernigan-Lin bisection (15); for an overview of properties relevant to the present article, see Table 1. We used the following implementations of the algorithms:

- WGCNA: blockwiseModules function (version 1.66) (16)
- MCL: markov_clustering (version 0.0.5) (https://github.com/GuyAllard/markov_clustering)
- Louvain method: python-louvain (version 0.11) (https://github.com/taynaud/python-louvain)
- Girvan-Newman algorithm: networkx (version 2.1)
- Kernighan-Lin bisection: networkx (version 2.1)

We supplied both the complete network and the positive-edge-only network to each algorithm except WGCNA and manta. WGCNA received the simulated data set instead of the Pearson correlations. For its evaluation, only assigned nodes were considered (i.e., nodes with poor correlations to cluster eigenvectors were ignored). We also tested scaled correlations (see Fig. S9 in the supplemental material). A range of parameter settings was tested for manta, the Louvain method, the Kernighan-Lin algorithm, and MCL (Fig. S2 to S5). We set these parameters as follows:

- manta: ratio set to 0.8, edgescale to 0.3
- MCL: on complete networks, inflation was set to 3 and expansion to 15; on positive-edge-only networks, inflation was set to 2 and expansion to 7
- Louvain method: on complete networks, resolution was set to 0.1; on positive-edge-only networks, resolution was set to 1
- Girvan-Newman algorithm: no parameter settings
- Kernighan-Lin bisection: on all networks, max_iter was set to 10

Cluster assignments were evaluated as described previously (24). The true positives were constructed through K-means clustering for the gLV approach. For the FABIA approach, identical indices were used to generate vectors with activated genes, so the true positives could be extracted from these indices. We report both the complex-wise sensitivity (Sn), the cluster-wise positive predictive value (PPV), geometrical accuracy (Acc), and the separation (Sep) (Fig. S1). We also report the sparsity score (equation 1). When a cluster size exceeded 80% of the total number of assigned species, all measures except the sparsity score were replaced with missing values to avoid obfuscation of the results. The opposite case—when nearly all species were assigned to their own clusters—was also replaced when the number of assigned clusters exceeded 50.

**Cheese rind case study.** We downloaded BIOM-formatted bacterial abundances of this cheese data set (study no. 11488 [25]) from Qiita (45). The data set was preprocessed by removing samples with fewer than 10,000 counts, rarefying them to an even depth, and filtering taxa with less than 20% prevalence. The final data set therefore included 97 taxa and 337 samples. We used these abundances to construct a network with CoNet; an initial multigraph was constructed with Pearson correlation, Spearman correlation, mutual information, Bray-Curtis dissimilarity, and Kullback-Leibler dissimilarity. Only edges that were supported by at least two methods were retained. The bootstrapping procedure further removed edges with $P$ values below 0.05, where $P$ values were merged by Brown's method and the Benjamini-Hochberg correction for multiple testing was applied. For clustering with manta, edge weights were converted to $-1$ or 1 based on the inferred sign. Additionally, we inferred Spearman correlations of taxon abundances to moisture, where taxon abundances were either summed per phylum or per cluster. manta was run with default settings (edge scale and ratio set to 0.8).

**Longitudinal coastal plankton case study.** We downloaded the supplementary files from reference 26. Taxon abundances were first agglomerated at the genus level; then, we removed samples with fewer than 2,000 counts and rarefied them to an even depth. Taxa with a sample prevalence below 30% were removed, with the total counts preserved by including them in a bin. This resulted in a final data set of 143 taxa and 260 samples. Removed samples were given as columns with NA values, and the time series supplied to eLSA therefore contained 90 time points with 3 replicates per time point. We then ran eLSA (v1.0.2; Python v2.7.12) with simple replicate merging (averaging replicates) and a delay of 5 (27, 28). For downstream analysis with manta, only associations with $P$ values below 0.05 and $q$ values below 0.05 were taken into account. The network was treated as undirected, and edge weights were converted to $-1$ and 1 prior to clustering. manta was run with default settings except for an edge scale of 0.2; with the default edge scale, nearly all nodes in the network were determined to be weakly assigned.

For an additional analysis of the clusters, a principal-coordinate analysis of the metadata with Bray-Curtis dissimilarity was carried out. Technical replicates were averaged, and the metadata features supplied by reference 26 were used to fit environmental vectors onto the ordination. The significance of covariation of these vectors with community composition was assessed through permutation testing (1,000 permutations), and only vectors with a $P$ value below 0.05 (after Benjamini-Hochberg multiple testing correction) and a $q$ value below 0.05 were retained. To compare the directions of cluster abundances to these vectors, all bacterial abundances were summed and the covariance between these abundance vectors and principal-component vectors was estimated. These covariance values significantly exceeded covariances of permuted bacterial abundances (1,000 permutations, $P$ value of 0.001) and were used to construct cluster abundance vectors; an arbitrary scaling coefficient was used for visualization purposes.

**Data availability.** All code for manta is available under the Apache-2.0 license at https://github.com/ramellose/manta. An archived version of manta, together with code and synthetic data used for writing the paper, is available from https://doi.org/10.5281/zenodo.3578106.

## SUPPLEMENTAL MATERIAL

Supplemental material is available online only.
**FIG S1**, PDF file, 0.04 MB.
**FIG S2**, PDF file, 0.5 MB.
**FIG S3**, PDF file, 0.4 MB.
**FIG S4**, PDF file, 0.2 MB.
**FIG S5**, PDF file, 0.3 MB.
**FIG S6**, PDF file, 0.1 MB.
**FIG S7**, PDF file, 0.03 MB.
**FIG S8**, PDF file, 0.1 MB.
**FIG S9**, PDF file, 0.1 MB.

## ACKNOWLEDGMENTS

We acknowledge Didier Gonze for helpful feedback on the balance of signed matrices and Jan Aerts for pointing out the "enemy's enemy is my friend" principle.

This project was supported by the KU Leuven under grant no. STG/16/006. K.F. received funding from the European Research Council (ERC) under the European Union's Horizon 2020 research and innovation program under grant agreement no. 801747.

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
