## [Reviewer comments · mSystems]

***manta* - a clustering algorithm for weighted ecological networks.**

Lisa Röttjers and Karoline Faust

Corresponding Author(s): Karoline Faust, KU Leuven

Review Timeline:

Submission Date:

December 20, 2019

Accepted:

January 28, 2020

Editor: Morgan Langille

Reviewer(s): The reviewers have opted to remain anonymous.

Transaction Report:

DOI: <https://doi.org/10.1128/mSystems.00903-19>

Reviewer #1 (Comments for the Author):

Manuscript:

mSystems00688-19

manta - a clustering algorithm for weighted ecological networks, by Lisa Röttjers and Karoline Faust

Summary:

This manuscript introduces a network clustering algorithm called "manta", which groups together nodes based on both edge density and [signed] weights. The method is based on a modification of the popular MCL algorithm and, although quite general, the authors apply manta to simulated and real microbiome data. This demonstrates the utility of the tool in relating manta-detected clusters to key questions in microbial ecology experiments.

For example, CoNet clusters identified by manta corresponds to moisture-tolerating species in a public cheese-rind 16S dataset - in the original paper only broad-correlations with moisture could be detected.

For situations where the core algorithm cannot stably converge, manta is run on graph subsets and the solutions are combined.

General comments:

This manuscript presents a novel clustering algorithm that addresses the task of clustering nodes in a graph in a way that is uniquely suited to microbial ecology. By explicitly accounting for the signs of taxa-taxa correlations in a modified Markov clustering algorithm, simulated network flows can partition nodes into anti-correlated groups.

The application to real-data experiments are convincing and the interpretations are reasonable. Also, "manta" is a great name for the software package!

The authors are forward looking, but are clear in how their tool should be correctly applied. The Discussion clearly lays out the use-case and when application of other tools would be more appropriate (e.g. if too few negative edges are available).

We thank the reviewer for these positive comments and for her/his thoughtful feedback.

However, the way the method is introduced and explained is confusing. The organization of the manuscript itself hampers clear communication. For example, I didn't understand what different clustering objective manta was even using until the Discussion. Furthermore, terms for mathematical operations or metrics, are occasionally used without being formally defined.

While technical details are rightly reserved for the Methods, as this section is at the end of paper a linear reading will leave even savvy readers without a thorough understanding of the method. I think that a brief explanation should also be included in the Results.

My suggestion is to reorganize the manuscript. The principle of "the enemy of my enemy is

my friend" is the key insight and should be brought into the introduction, and motivated by real examples from microbial ecology. Then it will be clear why existing tools are insufficient for this task. Figure 5 could be moved to the first figure. The authors may want a visual description of random walks along network edges ("network flow" is never defined).

As suggested by the reviewer, we have reorganized the manuscript to better present the algorithm. We agree that a linear reading would not give readers a good understanding of the algorithm. Therefore, we made several changes to the introduction and the results section.

In the introduction, we have chosen to include the following paragraph:

Page 2, line 65:

“Although some of these clustering approaches can cluster weighted networks, they have not specifically been designed to detect association patterns generated through ecological processes. Even if bacteria are not directly interacting, they may co-occur as a result of niche filtering (Gibbons et al. 2017), or they may actively prevent establishment of other species. Such antagonistic relationships have been described for sponge symbionts (Blunt et al. 2011), the plant rhizosphere (Durán *et al.* 2018) and biofilm formation (Ren et al. 2014; Oliveira *et al.* 2015). Regardless of the ongoing process, co-occurrence with one species but not another can be described as 'the enemy of my enemy is my friend' in an undirected network. Even if there are no direct interactions between co-occurring species, a powerful shared negative association introduced through niche filtering or competition can be sufficient to result in observed co-occurrence.”

Additionally, we added a brief summary of the method to the start of the Results section. We chose to omit 'network flow' and adapted Figure 5 (now Figure 1) to show how matrix multiplication involves walks from one part of the network. We clearly define random walks in the Methods section and contrast this to our approach.

Page 2, line 87:

“*manta* implements features for clustering of weighted and noisy networks

The indirect effect of one node on another node in the network can be estimated by multiplying the edges connecting the two nodes (Fig. 1A). If the nodes are only connected by positively-weighted edges, the indirect effect is also positive. In contrast, if the path between the two nodes contains a single negatively-weighted edge, the indirect effect is negative; hence, clusters found by *manta* reflect the principle 'the enemy of my enemy is my friend'. (Fig. 1A, B), *manta* uses two alternative strategies to generate scoring matrices (Fig. 1C, D). We refer to the Methods section for a detailed explanation and the pseudocode describing the algorithm.

After the scoring matrix is generated, it can be clustered with an agglomerative clustering approach (Fig. 1E). The optimal cluster number is identified with the sparsity score (Equation 1 in Methods), which is calculated from intra- and inter-cluster weighted edges. The network can then be rewired and the procedure repeated to generate robustness scores (Fig. 1F). This approach generates biologically relevant clusters while ignoring nodes that cannot be confidently assigned to a cluster.”

I also have some questions/suggestions for presenting the method itself:

It's clear that the scoring matrix is not the same stochastic matrix as in MCL. Here, since the signs are preserved and weights are scaled by the largest values (rather than the sum), network weights are not probabilities and so *manta* is not *exactly* modeling random walks. However, the manuscript is never very explicit about what process *_is_* actually occurring during

simulated "network flow": details on the algorithm, and including an analysis of algorithm complexity, would be appreciated here.

I also think presenting the graph traversal section as an algorithm in pseudo-code would provide a more compact and clear description. Each matrix update could use iteration-based indexing notation (M^i, M^{i+1}, \dots) and/or matrix element indexing, rather than introducing additional parameters m, I . Then each Inflation, Normalization, Expansion, Evaluation step can be clearly labelled with respect to an iteration.

As suggested by the reviewer, we have thoroughly rewritten the methods section so it more clearly describes the algorithm.

Firstly, we rewrote the initial paragraph to appropriately describe why *manta* is not modelling Markov chains (a specific subset of random walks). *manta* is modelling random walks, as the matrix power is the expected outcome of all random walks across the graph. However, it is not modelling random walks as probabilities, but as path products. We also omitted all references to 'network flow', since this has a precise definition (flow is the capacity of an edge in a flow network to handle a specific amount, e.g. a flow of cars on a road) that does not exactly apply to the *manta* algorithm.

Page 13, line 313:

“We can represent a graph as an adjacency matrix, with each node in the graph represented as a row and column in the matrix. A non-zero entry in row i and column j represents an edge between node i and node j . In the case of an undirected graph, the adjacency matrix is symmetric, so position i, j is identical to position j, i . The weighted adjacency matrix, where the values in the matrix are the edge weights, is used by *manta* to cluster networks without losing sign information.

manta was originally designed as an alternative to MCL, which uses random walks to find clusters. MCL does so through iterations of matrix expansion and inflation so that values in the matrix eventually converge to 0 or 1. The process of modelling random walks as Markov chains requires the adjacency matrix to be converted to a stochastic matrix in each iteration, so that all columns sum to one. This normalization step removes the signs from all values. Since *manta* uses an alternative normalization method to retain the signs, *manta* does not model Markov chains. The pseudocode below describes the core algorithm, with line 7 describing both the expansion and subsequent normalization and line 9 the inflation step (Algorithm 1).”

Secondly, we added pseudocode to describe two important versions of the core algorithm: a shorter algorithm that is used on balanced graphs, and a longer algorithm that is used on unbalanced graphs.

We provide two screenshots of the pseudocode below.

Algorithm 1 Pseudocode for normalizing scoring matrix

```
1  $M \leftarrow$  Scoring matrix
2 for row, column in  $M$  do
3    $M_{row,column} \leftarrow M_{row,column} / \max(\text{abs}(M))$ 
4 end
```

Algorithm 2 Pseudocode for balanced graphs

```
1  $M[0] \leftarrow$  Scoring matrix initialized from weighted adjacency matrix
2  $\epsilon \leftarrow$  Threshold for convergence
3  $E \leftarrow$  Initial error value, set to 1
4  $I \leftarrow$  Maximum number of iterations
5 for  $i = 1$  to  $I$  do
6   while  $100E > \epsilon$  then
7      $M[i] \leftarrow M[i-1]^2$ 
8      $M[i] \leftarrow$  apply algorithm 1 on  $M[i]$ 
9     for row, column in  $M[i]$  do
10       $M[i]_{row,column} \leftarrow M[i]_{row,column} + 1 / M[i]_{row,column}$ 
11    end
12     $M[i] \leftarrow$  apply algorithm 1 on  $M[i]$ 
13     $E \leftarrow 0$ 
14     $b \leftarrow 0$ 
15    for row, column in  $M[i]$  do
16      if  $M[i]_{row,column}$  is not 0 then
17         $b \leftarrow b + 1$ 
18         $E \leftarrow M[i]_{row,column} - M[i-1]_{row,column}$ 
19      end
20     $E \leftarrow E / b$ 
21 end
```

Algorithm 3 Pseudocode for unbalanced graphs

```
1   $G \leftarrow$  Original graph
2   $K \leftarrow$  Number of subgraphs to generate
3   $F \leftarrow$  Number of edges to subsample
4   $R \leftarrow$  Fraction of subgraphs where edges need to be stable
5  for  $k = 1$  to  $K$  do
6      sample  $F$  edges from  $G$  to create  $g$ 
7      if components in  $g$  are balanced then
8          for component in  $g$  do
9               $B \leftarrow$  apply algorithm 2 on component
10         end
11      $M[k] \leftarrow$  Scoring matrix initialized from  $g$ 
12      $M[k] \leftarrow M[k]^2$ 
13      $M[k] \leftarrow$  apply algorithm 1 on  $M[k]$ 
14     for row, column in  $M[k]$  do
15          $M[k]_{row,column} \leftarrow M[k]_{row,column} + 1/M[k]_{row,column}$ 
16     end
17      $M[k] \leftarrow$  apply algorithm 1 on  $M[k]$ 
18      $M[k] \leftarrow M[k] + B$ 
19      $M[k] \leftarrow$  apply algorithm 1 on  $M[k]$ 
20 end
21  $O \leftarrow$  empty matrix
22 for row, column in  $M$  do
23      $p \leftarrow 0$ 
24      $n \leftarrow 0$ 
25     for  $k = 1$  to  $K$  do
26         if  $M[k]_{row,column} > 0$  then
27              $p \leftarrow p + 1$ 
28         else if  $M[k]_{row,column} < 0$  then
29              $n \leftarrow n + 1$ 
30     if  $p/K > R$  or if  $n/K > R$  then
31         for  $k = 1$  to  $K$  do
32              $O_{row,column} \leftarrow O_{row,column} + M[k]_{row,column}$ 
33         end
34 end
35  $O \leftarrow O/\max(\text{abs}(O))$ 
```

We also included a brief analysis of algorithm complexity:

Page 13, line 338:

“Like MCL, the time and space complexity of *manta* is a function of the matrix dimensions. The most time-intensive operation is the matrix power: for a $n \times n$ matrix, the time complexity of a naïve matrix power is $O(n^3)$ (Pan 1984). MCL has tackled this problem through a pruning strategy that removes values close to zero, which reduces the time complexity of the

algorithm. However, the complexity of the *manta* intermediate matrices increases rather than decreases as each position converges towards -1 or 1 , so no pruning strategy could be implemented. Since *manta* uses numpy to carry out matrix multiplications, the time complexity is instead determined by high-performance BLAS implementations and is a bit lower than $O(n^3)$ in practice (Dongarra *et al.* 1988).

As the entire matrix is stored in memory, the space complexity is $O(n^2)$. The time and space complexities provided above ignore the number of iterations; generally, between 50 and 100 iterations are sufficient to generate a scoring matrix for an unbalanced graph, while a balanced graph will need even fewer. On the 100-species gLV example, the complete algorithm took approximately 7 seconds to run (Windows, Intel Xeon E3-1505M at 3.00 Ghz, 32 GB memory).”

Finally, for simulated data it's not quite clear that the simulation was constructed such that a pre-determined clustering-solution is built into the network. It seems that "sparsity score" should be maximized and the separation should be 0.5 (but only for two cluster networks?), but these metrics seem independent of a ground truth clustering solution in some simulation setups.

We agree with the reviewer that the ground truth was not defined clearly.

In our gLV implementation, we chose to add an environmental perturbation vector that resulted in the appearance of clusters. We used this perturbation vector as a ground truth. However, given that there are strong interactions in the gLV simulation, many species that should do poorly in one condition still grow; they are rescued through positive interactions with others. The opposite is also true. Consequently, our ground truth did not reflect the actual abundance patterns, and the separation could only reach about 0.5.

An alternative option would be to define clusters on the matrix level, so no environmental perturbations are necessary and the clusters will instead reflect the interactions. However, this is not straightforward with gLV simulations, as the high number of positive interactions cause the interaction matrix to become unstable.

Given that the perturbation vector does not accurately reflect the ground truth and clustering through the interaction matrix is not straightforward, we chose to use K-means clustering on the dataset to define a ground truth. This ground truth contains both the effect of the interaction network and the environmental perturbation instead of just the perturbation. As a result, the separation can approach 1 for some algorithms. However, this simpler evaluation strategy also implies that a clustering approach on the abundance matrix can give better results than the network clustering.

In the FABIA simulation, the boundaries were not defined clearly in terms of activated taxa (e.g. abundance vectors with added noise). We re-defined these boundaries and simplified the code so the true positives better matched the observed networks.

This has been clarified in the Methods:

Page 18, line 501:

“Cluster assignments were evaluated as described by (Brohee 2006). The true positives were constructed through K-means clustering for the gLV approach. For the FABIA approach, identical indices were used to generate vectors with activated genes, so the true positives could be extracted from these indices.”

We added a brief explanation of the sparsity score in the start of the Results section and refer to the corresponding section in the Methods:

Page 4, line 97:

“After the scoring matrix is generated, it can be clustered with an agglomerative clustering approach (Fig. 1E). The optimal cluster number is identified with the sparsity score (Equation 1 in Methods), which is calculated from intra- and inter-cluster weighted edges. The network can then be rewired and the procedure repeated to generate robustness scores (Fig. 1F). This approach generates biologically relevant clusters while ignoring nodes that cannot be confidently assigned to a cluster.

Page 4, line 118:

“Additionally, we included the sparsity score used by *manta* (Equation 1) to visualize the ratio of intra- and inter-cluster weighed edges.”

I think overall the methods and applications are interesting, but to have the biggest impact, the authors should rework the manuscript to present details such that they are accessible and clear to most readers.

Specific Comments:

P2,L48 “..MCL uses random walks based on integers...”

This definition should be clarified - in what way is MCL based on integers? I suppose you can assume readers now what integers are, but knowing that whole numbers are somehow involved in MCL will not help readers evaluate it nor set up a meaningful contrast with *manta*.

P2,L49 “..scaling values or by adjusting the inflation parameter..”:

None of these terms are defined, so again this is not a useful description of MCL.

We changed the sentence as follows and included a detailed explanation of random walks in the methods:

Page 2, line 50:

“For example, the Markov Cluster Algorithm (MCL) uses a probability matrix to identify clusters (Van Dongen 2000). While a weighted adjacency matrix can be scaled to generate a probability matrix, the algorithm depends on edge density to infer clusters and is therefore mostly suitable for networks with a low number of negatively-weighted inter-cluster edges.”

P2,L65 “...error rates...”

Readers may not know that statistical network inference tools are prone to false positives, so some clarification might be useful here.

We introduced a reference in an earlier paragraph:

Page 2, line 44:

“Unlike many ecological networks, microbial association networks suffer from interpretational challenges as predicted interactions cannot be observed directly (Röttgers and Faust 2018). Moreover, tools used to infer associations generally suffer from high error rates (Weiss *et al.* 2016).”

P2,L77 “achieve a separation around 0.5”

The goal for clustering synthetic networks is insufficiently explained and “separation” is not defined, so it's not clear if this is a good result.

We added the following paragraphs to the Results section:

Page 4, line 116:

Cluster assignments were evaluated with the complex-wise sensitivity (S_n), the cluster-wise positive predictive value (PPV), geometrical accuracy (Acc) and the separation (Sep) (Brohee

and Van Helden 2006). Additionally, we included the sparsity score used by *manta* (Equation 1) to visualize the ratio of intra- and inter-cluster weighed edges. The complex-wise sensitivity estimates the coverage of a true positive cluster by its best-matching assigned cluster, whereas the cluster-wise positive predictive value measures how well an assigned cluster covers its best-matching true positive cluster. In contrast, the separation is calculated by taking the product of the fraction of assigned nodes in the true positive clusters by the fraction of true-positive nodes in the assigned clusters. Hence, the separation penalizes for cluster overlap, unlike the reported Acc, PPV and Sn.

This approach uses the contingency matrix rather than a list of true positives, effectively permitting evaluation of assignments that do not necessarily match the true positive clusters (Fig. S1). However, these measures can be skewed by cluster assignments that mostly assign all nodes to one cluster (Fig. S1B) or assign almost every node to its own cluster.

Additionally, overlapping clusters can inflate some measures of performance (Fig. S1C). To resolve these pathological cases, we filtered assignments where over 80% of the nodes were assigned to a single cluster, or over 50 clusters were identified.

While Acc, PPV and Sn can be high for algorithms that assign true positive clusters to a single cluster, separation is calculated by multiplying the proportion of true-positive nodes in the assigned cluster with the proportion of cluster nodes in the true-positive cluster. Hence, the separation measure punishes cluster assignments that mix up multiple true-positive clusters.”

P5,L97-98, P15,L417-420

A few things are unclear w.r.t to FABIA simulations. Firstly, what data are you actually simulating? Biclustering is performed on the sample by feature data matrix, so what in the network input to *manta*? I'm assuming that it's a correlation matrix from the full set of observations. If so, it's not clear to me that biclusters should generate condition-coherent node clusters, since the correlation signal might not be strong enough to be detected through uncorrelated data subsets.

We used the FABIA code to generate the sample by feature (=taxon) data matrix used to evaluate biclustering strategies. From this matrix, we constructed a Pearson correlation network. We only permitted a small number of features (20 out of 100) to be uncorrelated, so the biclusters generated condition-coherent clusters. We included a heatmap of a generated

dataset below:

We have adjusted the Methods section to better reflect the network inference method:
Page 17, line 468:

“The second type of simulation uses an adapted function from the FABIA R package (Hochreiter *et al.* 2010), a package that generates simulated data for evaluations of biclustering algorithms. We adapted the `makeFabiaDataBlocksPos` function to generate biclusters at set locations and used these locations to define true positive clusters. The constructed matrix was then used to infer a correlation network in an identical manner as the gLV networks.”

Furthermore, if there is no underlying ground-truth association network, what does even mean that a clustering an inferred network could lead to a "correct" clustering? It seems like a biologist would get (better?) results from just biclustering the data matrix, rather than clustering a network.

If the authors want to encode environmental covariates into the simulation, perhaps a mixed graphical model approach would fall more cleanly into the network inference framework. Otherwise I think the use of FABIA biclustering needs stronger justification.

Concerning the absence of a “correct” clustering solution: The ground truth is meaningfully defined also in the absence of an association network through membership in co-varying blocks. We agree that biclustering on the data matrix would probably give better results than the network approach in this particular noise-free simulation. Our main purpose was to demonstrate that, in absence of any network structure, *manta* can still return sensible results. If an underlying interaction network is missing, the co-occurrence patterns may represent outcomes of niche filtering or neutral dynamics (with slow migration). An association network can visualize such features despite the absence of an underlying interaction network.

We added a more detailed justification in the Results section:
Page 4, line 105:

“To evaluate performance of *manta* compared to alternative methods, we generated synthetic

data sets using two different approaches. One is based on the generalized Lotka-Volterra (gLV) equation, while the other (FABIA) was developed for the evaluation of biclustering applied to gene expression data (Hochreiter 2010). We chose to use these approaches because they provide a ground truth and have entirely different network topology. For example, the median approximated node connectivity of the gLV networks is 1, in contrast to 46 for the FABIA networks. The latter networks more closely represent ecosystems without biotic interactions, where network topology is governed solely by niche filtering or other dynamics.”

P6,L105

The "sparsity" score has not been defined yet.

We added a reference to the appropriate equation:

Page 4, line 118: “Additionally, we included the sparsity score used by *manta* (Equation 1) to visualize the ratio of intra- and inter-cluster weighed edges.”

P7,Fig3B

PCoA is missing a color legend

We have added a color legend for the cheese rind types.

P8,L154:

Something like a network clustering-driven approach to identifying important subcompositions is an interesting application that don't think I've seen before. Perhaps highlight this possible novel application?

We included the following sentence in the introduction:

Page 2, line 78:

“We demonstrate in two case studies that our method can identify subcompositions with biological importance that cannot be found through conventional correlation strategies.”

P11,L279-280 "reversed differences rather than reinforcing them"

This is ambiguous - differences between what?

We clarified the sentence as follows:

Page 13, line 331:

“While this step may seem counter-intuitive as it reduces the difference between small and large values in the matrix (rather than inflating them as in MCL), values converge to -1 and 1 each time for the toy model (Fig. 1B).”

P11,L284 M after inflation before normalization

Perhaps additional representation should be used for this matrix, along with an iteration index so that there can be no confusion about the value.

As suggested, we have replaced these equations with pseudocode.

P12,Fig5A

Perhaps nodes can be labelled/numbered so that nodes can be visually mapped to entries in the adjacency/scoring matrices and clustering vectors?

We have simplified the toy model to include the node labels without overcrowding the figure. Please find the updated figure below.

Fig5B

A color legend for weights would be helpful here. It seems that black squares correspond to negative edges? Also, perhaps each matrix should be labelled 1-6 to indicate each step in the

iteration.

We have updated the figure in accordance with these suggestions.

Fig5C

The flow chart from the toy network to the matrices in 5C is misleading, since the 100x100 matrices here don't correspond to the 10-node network. Additionally It's not that clear from this figure what a flip-flop state actually is. If it means the algorithm isn't converging or that two runs result in different end states perhaps this can be shown by continuing the iteration. This could also be a good place to show what an unbalanced cycle looks like graphically (perhaps in a subset of the 100x100 matrix).

Indeed, the flowchart incorrectly suggests that the 100x100 matrix corresponds to the toy model. Therefore, we have dropped the flowchart entirely in favour of a much smaller unbalanced toy graph and its associated iterations.

FIG 1 *manta* pipeline. (A) Toy graph with two clusters separated by negatively-weighted edges. The effect of node x on node z can be estimated by taking the product of edges $1, 2$ and $2, 5$. (B) Toy graph with a single negatively-weighted edge in the left cluster. (C) Scoring matrix for (A) across six iterations. Black and white values reflect -1 and 1 respectively. After six iterations, the scoring matrix reaches convergence. (D) Scoring matrix for (B) across nine iterations. Unlike (C), this matrix reaches a flip-flop state, where the scoring matrix alternates between the configurations shown in iterations 6, 7, 8 and 9. A few values in the matrix reach -1 or 1 while all other values oscillate near 0 . (E) *manta* uses agglomerative clustering on the scoring matrix to assign each node to a cluster. For flip-flopping matrices, the scoring matrix is generated from subsets of the complete network. (F) A fraction of the original network is rewired to generate permuted cluster assignments with identical degree distributions. Robustness of cluster assignments can then be estimated by comparing the Jaccard similarity of cluster memberships cluster-wise or node-wise.

P13,L302

The indexing notation C_i is later dropped in equation 4, or perhaps C (with no index) is actually the set of all edges in any cluster?

We removed the indexing notation from the main text, as only the total set of edges inside clusters is relevant.

P13,L323

The authors could be a bit more explicit with their graph theory. For example, define cycle so that the idea of balanced graphs is clear (and an figure might help too).

We added a more interpretable definition and refer to Figure 1A-B:

Page 14, line 360:

“A cycle is a closed chain of nodes; for example, the cycle 1, 2, 3 in the balanced toy model (Fig. 1A) has a positive product, while the same cycle in the unbalanced toy model has a negative product.”

P13,L329 "previously described diffusion procedure..."

Ambiguous to where this was previously described: is there a citation missing?

Also, a fuller definition of belief propagation in this context would be useful. Is the general idea to combine manta solutions on subgraphs that are themselves balanced? Can you provide any convergence guarantees in the context of manta or, if that's out of scope, by analogy with previous work on MCL-like algorithms?

Obviously this isn't meant to be a CS paper, but some a more comprehensive algorithm analysis would probably help the method be adopted in other fields.

We have chosen to rewrite this paragraph to reduce the complexity and added pseudocode for both versions of the algorithm (Algorithm 1 and Algorithm 2). We hope this clarifies the methodology. Balanced subgraphs appear to be rare, but they are indeed included in Algorithm 3.

New version:

Page 14, line 354:

“As described by Van Dongen (Van Dongen 2000), classes of matrices exist that do not converge to a stable state after repeated iterations. Instead, these matrices exhibit flip-flop equilibrium states and switch to alternative configurations with each iteration.

While these flip-flop states represent rare cases when MCL is applied, the use of signed graphs by *manta* strongly increases the probability of these states appearing.

This relates to the notion of balance in signed graphs (Harary 1953), where a graph is only balanced if the product of edge weights in every cycle is positive. A cycle is a closed chain of nodes; for example, the cycle 1, 2, 3 in the balanced toy model (Fig. 1A) has a positive product, while the same cycle in the unbalanced toy model (Fig. 1B) has a negative product.

The balance of the graph matters for the expansion step; we observed that the sign of non-zero elements of the expanded scoring matrix never conflicts with the sign of non-zero elements in the weighted adjacency matrix if the adjacency matrix corresponds to a balanced graph.

If the graph is not balanced (Harary and Kabell 1980), *manta* carries out the previously described diffusion procedure on a subset of nodes (Algorithm 2). However, only one iteration is carried out on this subset, as any more iterations would lead to the appearance of flip-flop states. The subsetting approach is complemented by an analysis of any balanced components in the graph; if those are present, multiple iterations of expansion and inflation are carried out on the balanced subgraph until convergence occurs. The scoring matrix used

for cluster assignment is reconstructed from the subsets, but only from positions in the subsets where the sign of the value is consistent with the sign reported by most subsets. However, the accumulation of high scores for central nodes prevents the agglomerative clustering algorithm from identifying relevant clusters. Therefore, whenever small clusters with a size below a user-specified threshold are detected, rows and columns in the scoring matrix that correspond to these small clusters are removed and clusters are calculated from the remaining values. The removed nodes are then assigned to a cluster based on the average shortest path weights to cluster members.”

Unfortunately, we were not able to provide mathematical proof of convergence. While there is a large body of work on stochastic matrices and random walks, the input graph for *manta* can does not describe a Markov process. Given the differences between the MCL algorithm and *manta*, a formal mathematical analysis is beyond the scope of this work and beyond our expertise. However, we have never observed an absence of convergence when Algorithm 3 is applied, as this algorithm does not require the matrix to fully converge to -1 and 1.

P14,L353 "a few diagonal values..."

Are diagonal elements non-zero, in general, when the graph is balanced?

Yes, if the graph is balanced, all diagonal values become positive.

Page 16, line 405:

“The limits of the scoring matrix approach -1 and 1 during flip-flop iterations, but these limits are only approached by a few diagonal values; in a balanced matrix, all diagonal values should approach 1 (Fig. 1C).”

P15,L434

Have you considered also supplying scaled correlation distances to the clustering algorithms as well (as opposed to positive-only) so that edges aren't missing?

We have indeed supplied algorithms with scaled correlation distances, but this significantly reduced performance of most algorithms that did well on the positive-edge-only networks. We have included the result of this analysis in the supplements (Figure S9) and updated the Methods to refer to this figure.

Page 17, line 486:

“We supplied both the complete network and the positive-edge only network to each algorithm except WGCNA and *manta*. WGCNA received the simulated data set instead of the Pearson correlations. For its evaluation, only assigned nodes were considered (i.e. nodes with poor correlations to cluster eigenvectors were ignored). We also tested scaled correlations (Fig. S9).”

FIG S9 Performance of clustering algorithms on networks with the range of edge weights shifted to 0 and 1.

Sensitivity (Sn), positive predictive values (PPV), accuracy (Acc) and separation (Sep) were calculated as described by (Brohee and Van Helden 2006). Sparsity of the assignment is a function of the edge weights of intra-cluster versus inter-cluster edges (Equation 1). Clustering performance was estimated on 50 independently generated datasets. Matrices of taxon abundances were generated from a synthetic random interaction matrix. Clustering was carried out on Spearman correlation networks inferred from these matrices, with the range of correlations shifted to 0 and 1. As WGCNA constructed its own networks, shifting the edge weights was not possible for this tool and its performance therefore corresponds to performance on the normalized data. The numbers next to the sensitivity results indicate how many clustering assignments met the following criteria for a particular algorithm: no cluster should exceed 80% of the total number of nodes, and there should be fewer than 50 clusters.

- Blunt JW, Copp BR, Munro MHG *et al.* Marine natural products. *Nat Prod Rep* 2011;**28**:196–268.
- Brohee S, Van Helden J. Evaluation of clustering algorithms for protein-protein interaction networks. *BMC Bioinformatics* 2006;**7**:488.
- Dongarra JJ, Du Croz J, Hammarling S *et al.* An extended set of FORTRAN basic linear algebra subprograms. *ACM Trans Math Softw* 1988;**14**:1–17.
- Van Dongen S. A cluster algorithm for graphs. 2000.
- Durán P, Thiergart T, Garrido-Oter R *et al.* Microbial interkingdom interactions in roots promote Arabidopsis survival. *Cell* 2018;**175**:973–83.
- Gibbons SM, Kearney SM, Smillie CS *et al.* Two dynamic regimes in the human gut microbiome. *PLoS Comput Biol* 2017;**13**:e1005364.
- Harary F. On the notion of balance of a signed graph. *Michigan Math J* 1953;**2**:143–6.
- Harary F, Kabell JA. A simple algorithm to detect balance in signed graphs. *Math Soc Sci* 1980;**1**:131–6.
- Hochreiter S, Bodenhofer U, Heusel M *et al.* FABIA: factor analysis for bicluster acquisition. *Bioinformatics* 2010;**26**:1520–7.
- Oliveira NM, Martinez-Garcia E, Xavier J *et al.* Biofilm formation as a response to ecological competition. *PLoS Biol* 2015;**13**:e1002191.
- Pan V. How can we speed up matrix multiplication? *SIAM Rev* 1984;**26**:393–415.
- Ren D, Madsen JS, de la Cruz-Perera CI *et al.* High-throughput screening of multispecies biofilm formation and quantitative PCR-based assessment of individual species proportions, useful for exploring interspecific bacterial interactions. *Microb Ecol* 2014;**68**:146–54.
- Röttgers L, Faust K. From hairballs to hypotheses—biological insights from microbial networks. *FEMS Microbiol Rev* 2018.
- Weiss S, Van Treuren W, Lozupone C *et al.* Correlation detection strategies in microbial data sets vary widely in sensitivity and precision. *ISME J* 2016;**10**:1669–81.

January 28, 2020

Dr. Karoline Faust
KU Leuven
Department of Microbiology and Immunology
Herestraat 49
Leuven 3000
Belgium

Re: mSystems00903-19 (*manta* - a clustering algorithm for weighted ecological networks.)

Dear Dr. Karoline Faust:

Unfortunately, I was not able to solicit a review from the original referee. However, I have reviewed your responses to the initial reviewer as well as reviewed your revised manuscript. I have found your responses comprehensive and compelling and I think the manuscript has been greatly improved and provides a novel method of high interest for better understanding of microbial networks.

Therefore, my decision is that your manuscript be accepted, and I am forwarding it to the ASM Journals Department for publication. For your reference, ASM Journals' address is given below. Before it can be scheduled for publication, your manuscript will be checked by the mSystems senior production editor, Ellie Ghatineh, to make sure that all elements meet the technical requirements for publication. She will contact you if anything needs to be revised before copyediting and production can begin. Otherwise, you will be notified when your proofs are ready to be viewed.

Sincerely,

Morgan Langille
Editor, mSystems

Journals Department
Supplemental file 9: Accept
Supplemental file 6: Accept
Supplemental file 3: Accept
Supplemental file 2: Accept
Supplemental file 8: Accept
Supplemental file 4: Accept
Supplemental file 1: Accept
Supplemental file 5: Accept
Supplemental file 7: Accept